# MetaGrad: Multiple Learning Rates in Online Learning

**Tim van Erven**
Leiden University
tim@timvanerven.nl

**Wouter M. Koolen**
Centrum Wiskunde & Informatica
wmkoolen@cwi.nl

## Abstract

In online convex optimization it is well known that certain subclasses of objective functions are much easier than arbitrary convex functions. We are interested in designing adaptive methods that can automatically get fast rates in as many such subclasses as possible, without any manual tuning. Previous adaptive methods are able to interpolate between strongly convex and general convex functions. We present a new method, MetaGrad, that adapts to a much broader class of functions, including exp-concave and strongly convex functions, but also various types of stochastic and non-stochastic functions without any curvature. For instance, Meta-Grad can achieve logarithmic regret on the unregularized hinge loss, even though it has no curvature, if the data come from a favourable probability distribution. MetaGrad's main feature is that it simultaneously considers multiple learning rates. Unlike previous methods with provable regret guarantees, however, its learning rates are not monotonically decreasing over time and are not tuned based on a theoretically derived bound on the regret. Instead, they are weighted directly proportional to their empirical performance on the data using a tilted exponential weights master algorithm.

## 1 Introduction

Methods for *online convex optimization* (OCO) [28, 12] make it possible to optimize parameters sequentially, by processing convex functions in a streaming fashion. This is important in time series prediction where the data are inherently online; but it may also be convenient to process offline data sets sequentially, for instance if the data do not all fit into memory at the same time or if parameters need to be updated quickly when extra data become available.

The difficulty of an OCO task depends on the convex functions $f_1, f_2, \ldots, f_T$ that need to be optimized. The argument of these functions is a $d$-dimensional parameter vector $\boldsymbol{w}$ from a convex domain $\mathcal{U}$. Although this is abstracted away in the general framework, each function $f_t$ usually measures the loss of the parameters on an underlying example $(\boldsymbol{x}_t, y_t)$ in a machine learning task. For example, in classification $f_t$ might be the *hinge loss* $f_t(\boldsymbol{w}) = \max\{0, 1 - y_t \langle \boldsymbol{w}, \boldsymbol{x}_t \rangle\}$ or the *logistic loss* $f_t(\boldsymbol{w}) = \ln\left(1 + e^{-y_t \langle \boldsymbol{w}, \boldsymbol{x}_t \rangle}\right)$, with $y_t \in \{-1, +1\}$. Thus the difficulty depends both on the choice of loss and on the observed data.

There are different methods for OCO, depending on assumptions that can be made about the functions. The simplest and most commonly used strategy is *online gradient descent* (GD), which does not require any assumptions beyond convexity. GD updates parameters $\boldsymbol{w}_{t+1} = \boldsymbol{w}_t - \eta_t \nabla f_t(\boldsymbol{w}_t)$ by taking a step in the direction of the negative gradient, where the step size is determined by a parameter $\eta_t$ called the *learning rate*. For learning rates $\eta_t \propto 1/\sqrt{t}$, GD guarantees that the *regret* over $T$ rounds, which measures the difference in cumulative loss between the online iterates $\boldsymbol{w}_t$ and the best offline parameters $\boldsymbol{u}$, is bounded by $O(\sqrt{T})$ [33]. Alternatively, if it is known beforehand that the functions are of an easier type, then better regret rates are sometimes possible. For instance, if the

functions are *strongly convex*, then logarithmic regret $O(\ln T)$ can be achieved by GD with much smaller learning rates $\eta_t \propto 1/t$ [14], and, if they are *exp-concave*, then logarithmic regret $O(d \ln T)$ can be achieved by the *Online Newton Step* (ONS) algorithm [14].

This partitions OCO tasks into categories, leaving it to the user to choose the appropriate algorithm for their setting. Such a strict partition, apart from being a burden on the user, depends on an extensive cataloguing of all types of easier functions that might occur in practice. (See Section 3 for several ways in which the existing list of easy functions can be extended.) It also immediately raises the question of whether there are cases in between logarithmic and square-root regret (there are, see Theorem 3 in Section 3), and which algorithm to use then. And, third, it presents the problem that the appropriate algorithm might depend on (the distribution of) the data (again see Section 3), which makes it entirely impossible to select the right algorithm beforehand.

These issues motivate the development of *adaptive* methods, which are no worse than $O(\sqrt{T})$ for general convex functions, but also automatically take advantage of easier functions whenever possible. An important step in this direction are the adaptive GD algorithm of Bartlett, Hazan, and Rakhlin [2] and its proximal improvement by Do, Le, and Foo [8], which are able to interpolate between strongly convex and general convex functions if they are provided with a data-dependent strong convexity parameter in each round, and significantly outperform the main non-adaptive method (i.e. Pegasos, [29]) in the experiments of Do et al. Here we consider a significantly richer class of functions, which includes exp-concave functions, strongly convex functions, general convex functions that do not change between rounds (even if they have no curvature), and stochastic functions whose gradients satisfy the so-called Bernstein condition, which is well-known to enable fast rates in offline statistical learning [1, 10, 19]. The latter group can again include functions without curvature, like the unregularized hinge loss. All these cases are covered simultaneously by a new adaptive method we call *MetaGrad*, for multiple eta gradient algorithm. MetaGrad maintains a covariance matrix of size $d \times d$ where $d$ is the parameter dimension. In the remainder of the paper we call this version *full MetaGrad*. A reference implementation is available from [17]. We also design and analyze a faster approximation that only maintains the $d$ diagonal elements, called *diagonal MetaGrad*. Theorem 7 below implies the following:

**Theorem 1.** *Let $\boldsymbol{g}_t = \nabla f_t(\boldsymbol{w}_t)$ and $V_T^{\boldsymbol{u}} = \sum_{t=1}^{T} \left( (\boldsymbol{u} - \boldsymbol{w}_t)^{\intercal} \boldsymbol{g}_t \right)^2$. Then the regret of full MetaGrad is simultaneously bounded by $O(\sqrt{T \ln \ln T})$, and by*

$$\sum_{t=1}^{T} f(\boldsymbol{w}_t) - \sum_{t=1}^{T} f_t(\boldsymbol{u}) \ \leq \ \sum_{t=1}^{T} (\boldsymbol{w}_t - \boldsymbol{u})^{\intercal} \boldsymbol{g}_t \ \leq \ O\left( \sqrt{V_T^{\boldsymbol{u}} \, d \ln T} + d \ln T \right) \quad \textit{for any } \boldsymbol{u} \in \mathcal{U}. \ (1)$$

Theorem 1 bounds the regret in terms of a measure of variance $V_T^{\boldsymbol{u}}$ that depends on the distance of the algorithm's choices $\boldsymbol{w}_t$ to the optimum $\boldsymbol{u}$, and which, in favourable cases, may be significantly smaller than $T$. Intuitively, this happens, for instance, when there is stable optimum $\boldsymbol{u}$ that the algorithm's choices $\boldsymbol{w}_t$ converge to. Formal consequences are given in Section 3, which shows that this bound implies faster than $O(\sqrt{T})$ regret rates, often logarithmic in $T$, for all functions in the rich class mentioned above. In all cases the dependence on $T$ in the rates matches what we would expect based on related work in the literature, and in most cases the dependence on the dimension $d$ is also what we would expect. Only for strongly convex functions is there an extra factor $d$. It is an open question whether this is a fundamental obstacle for which an even more general adaptive method is needed, or whether it is an artefact of our analysis.

The main difficulty in achieving the regret guarantee from Theorem 1 is tuning a learning rate parameter $\eta$. In theory, $\eta$ should be roughly $1/\sqrt{V_T^{\boldsymbol{u}}}$, but this is not possible using any existing techniques, because the optimum $\boldsymbol{u}$ is unknown in advance, and tuning in terms of a uniform upper bound $\max_{\boldsymbol{u}} V_T^{\boldsymbol{u}}$ ruins all desired benefits. MetaGrad therefore runs multiple slave algorithms, each with a different learning rate, and combines them with a novel master algorithm that learns the empirically best learning rate for the OCO task in hand. The slaves are instances of exponential weights on the continuous parameters $\boldsymbol{u}$ with a suitable surrogate loss function, which in particular causes the exponential weights distributions to be multivariate Gaussians. For the full version of MetaGrad, the slaves are closely related to the ONS algorithm on the original losses, where each slave receives the master's gradients instead of its own. It is shown that $\lceil \frac{1}{2} \log_2 T \rceil + 1$ slaves suffice, which is at most 16 as long as $T \leq 10^9$, and therefore seems computationally acceptable. If not, then the number of slaves can be further reduced at the cost of slightly worse constants in the bound.

**Protocol 1:** Online Convex Optimization from First-order Information

---

**Input:** Convex set $\mathcal{U}$
 1: **for** $t = 1, 2, \ldots$ **do**
 2:    Learner plays $\boldsymbol{w}_t \in \mathcal{U}$
 3:    Environment reveals convex loss function $f_t : \mathcal{U} \to \mathbb{R}$
 4:    Learner incurs loss $f_t(\boldsymbol{w}_t)$ and observes (sub)gradient $\boldsymbol{g}_t = \nabla f_t(\boldsymbol{w}_t)$
 5: **end for**

---

**Related Work**   If we disregard computational efficiency, then the result of Theorem 1 can be achieved by finely discretizing the domain $\mathcal{U}$ and running the Squint algorithm for prediction with experts with each discretization point as an expert [16]. MetaGrad may therefore also be seen as a computationally efficient extension of Squint to the OCO setting.

Our focus in this work is on adapting to sequences of functions $f_t$ that are easier than general convex functions. A different direction in which faster rates are possible is by adapting to the domain $\mathcal{U}$. As we assume $\mathcal{U}$ to be fixed, we consider an upper bound $D$ on the norm of the optimum $\boldsymbol{u}$ to be known. In contrast, Orabona and Pál [24, 25] design methods that can adapt to the norm of $\boldsymbol{u}$. One may also look at the shape of $\mathcal{U}$. As can be seen in the analysis of the slaves, MetaGrad is based a spherical Gaussian prior on $\mathbb{R}^d$, which favours $\boldsymbol{u}$ with small $\ell_2$-norm. This is appropriate for $\mathcal{U}$ that are similar to the Euclidean ball, but less so if $\mathcal{U}$ is more like a box ($\ell_\infty$-ball). In this case, it would be better to run a copy of MetaGrad for each dimension separately, similarly to how the diagonal version of the AdaGrad algorithm [9, 21] may be interpreted as running a separate copy of GD with a separate learning rate for each dimension. AdaGrad further uses an adaptive tuning of the learning rates that is able to take advantage of sparse gradient vectors, as can happen on data with rarely observed features. We briefly compare to AdaGrad in some very simple simulations in Appendix A.1.

Another notion of adaptivity is explored in a series of work [13, 6, 31] obtaining tighter bounds for linear functions $f_t$ that vary little between rounds (as measured either by their deviation from the mean function or by successive differences). Such bounds imply super fast rates for optimizing a fixed linear function, but reduce to slow $O(\sqrt{T})$ rates in the other cases of easy functions that we consider. Finally, the way MetaGrad's slaves maintain a Gaussian distribution on parameters $\boldsymbol{u}$ is similar in spirit to AROW and related confidence weighted methods, as analyzed by Crammer, Kulesza, and Dredze [7] in the mistake bound model.

**Outline**   We start with the main definitions in the next section. Then Section 3 contains an extensive set of examples where Theorem 1 leads to fast rates, Section 4 presents the MetaGrad algorithm, and Section 5 provides the analysis leading to Theorem 7, which is a more detailed statement of Theorem 1 with an improved dependence on the dimension in some particular cases and with exact constants. The details of the proofs can be found in the appendix.

## 2   Setup

Let $\mathcal{U} \subseteq \mathbb{R}^d$ be a closed convex set, which we assume contains the origin $\boldsymbol{0}$ (if not, it can always be translated). We consider algorithms for Online Convex Optimization over $\mathcal{U}$, which operate according to the protocol displayed in Protocol 1. Let $\boldsymbol{w}_t \in \mathcal{U}$ be the iterate produced by the algorithm in round $t$, let $f_t : \mathcal{U} \to \mathbb{R}$ be the convex loss function produced by the environment and let $\boldsymbol{g}_t = \nabla f_t(\boldsymbol{w}_t)$ be the (sub)gradient, which is the feedback given to the algorithm.[1] We abbreviate the *regret* with respect to $\boldsymbol{u} \in \mathcal{U}$ as $R_T^{\boldsymbol{u}} = \sum_{t=1}^{T} (f_t(\boldsymbol{w}_t) - f_t(\boldsymbol{u}))$, and define our measure of variance as $V_T^{\boldsymbol{u}} = \sum_{t=1}^{T} \left( (\boldsymbol{u} - \boldsymbol{w}_t)^{\mathsf{T}} \boldsymbol{g}_t \right)^2$ for the full version of MetaGrad and $V_T^{\boldsymbol{u}} = \sum_{t=1}^{T} \sum_{i=1}^{d} (u_i - w_{t,i})^2 g_{t,i}^2$ for the diagonal version. By convexity of $f_t$, we always have $f_t(\boldsymbol{w}_t) - f_t(\boldsymbol{u}) \le (\boldsymbol{w}_t - \boldsymbol{u})^{\mathsf{T}} \boldsymbol{g}_t$. Defining $\tilde{R}_T^{\boldsymbol{u}} = \sum_{t=1}^{T} (\boldsymbol{w}_t - \boldsymbol{u})^{\mathsf{T}} \boldsymbol{g}_t$, this implies the first inequality in Theorem 1: $R_T^{\boldsymbol{u}} \le \tilde{R}_T^{\boldsymbol{u}}$. A stronger requirement than convexity is that a function $f$ is *exp-concave*, which (for exp-concavity parameter 1) means that $e^{-f}$ is concave. Finally, we impose the following standard boundedness assumptions, distinguishing between the full version of MetaGrad (left column) and the diagonal version (right

column): for all $\boldsymbol{u}, \boldsymbol{v} \in \mathcal{U}$, all dimensions $i$ and all times $t$,

$$
\begin{array}{ccc}
\text{full} & & \text{diag} \\
\|\boldsymbol{u} - \boldsymbol{v}\| \leq D^{\text{full}} & \qquad & |u_i - v_i| \leq D^{\text{diag}} \\
\|\boldsymbol{g}_t\| \leq G^{\text{full}} & & |g_{t,i}| \leq G^{\text{diag}}.
\end{array} \tag{2}
$$

Here, and throughout the paper, the norm of a vector (e.g. $\|\boldsymbol{g}_t\|$) will always refer to the $\ell_2$-norm. For the full version of MetaGrad, the Cauchy-Schwarz inequality further implies that $(\boldsymbol{u} - \boldsymbol{v})^{\intercal} \boldsymbol{g}_t \leq \|\boldsymbol{u} - \boldsymbol{v}\| \cdot \|\boldsymbol{g}_t\| \leq D^{\text{full}} G^{\text{full}}$.

## 3   Fast Rate Examples

In this section, we motivate our interest in the adaptive bound (1) by giving a series of examples in which it provides fast rates. These fast rates are all derived from two general sufficient conditions: one based on the directional derivative of the functions $f_t$ and one for stochastic gradients that satisfy the *Bernstein condition*, which is the standard condition for fast rates in off-line statistical learning. Simple simulations that illustrate the conditions are provided in Appendix A.1 and proofs are also postponed to Appendix A.

**Directional Derivative Condition**   In order to control the regret with respect to some point $\boldsymbol{u}$, the first condition requires a quadratic lower bound on the curvature of the functions $f_t$ in the direction of $\boldsymbol{u}$:

**Theorem 2.** *Suppose, for a given $\boldsymbol{u} \in \mathcal{U}$, there exist constants $a, b > 0$ such that the functions $f_t$ all satisfy*

$$
f_t(\boldsymbol{u}) \geq f_t(\boldsymbol{w}) + a(\boldsymbol{u} - \boldsymbol{w})^{\intercal} \nabla f_t(\boldsymbol{w}) + b\left((\boldsymbol{u} - \boldsymbol{w})^{\intercal} \nabla f_t(\boldsymbol{w})\right)^2 \qquad \text{for all } \boldsymbol{w} \in \mathcal{U}. \tag{3}
$$

*Then any method with regret bound* (1) *incurs logarithmic regret, $R_T^{\boldsymbol{u}} = O(d \ln T)$, with respect to $\boldsymbol{u}$.*

The case $a = 1$ of this condition was introduced by Hazan, Agarwal, and Kale [14], who show that it is satisfied for all $\boldsymbol{u} \in \mathcal{U}$ by exp-concave and strongly convex functions. The rate $O(d \ln T)$ is also what we would expect by summing the asymptotic offline rate obtained by ridge regression on the squared loss [30, Section 5.2], which is exp-concave. Our extension to $a > 1$ is technically a minor step, but it makes the condition much more liberal, because it may then also be satisfied by functions that do *not* have any curvature. For example, suppose that $f_t = f$ is a fixed convex function that does not change with $t$. Then, when $\boldsymbol{u}^* = \arg\min_{\boldsymbol{u}} f(\boldsymbol{u})$ is the offline minimizer, we have $(\boldsymbol{u}^* - \boldsymbol{w})^{\intercal} \nabla f(\boldsymbol{w}) \in [-G^{\text{full}} D^{\text{full}}, 0]$, so that

$$
f(\boldsymbol{u}^*) - f(\boldsymbol{w}) \geq (\boldsymbol{u}^* - \boldsymbol{w})^{\intercal} \nabla f(\boldsymbol{w}) \geq 2(\boldsymbol{u}^* - \boldsymbol{w})^{\intercal} \nabla f(\boldsymbol{w}) + \frac{1}{D^{\text{full}} G^{\text{full}}} \left((\boldsymbol{u}^* - \boldsymbol{w})^{\intercal} \nabla f(\boldsymbol{w})\right)^2,
$$

where the first inequality uses only convexity of $f$. Thus condition (3) is satisfied by *any fixed convex function*, even if it does not have any curvature at all, with $a = 2$ and $b = 1/(G^{\text{full}} D^{\text{full}})$.

**Bernstein Stochastic Gradients**   The possibility of getting fast rates even without any curvature is intriguing, because it goes beyond the usual strong convexity or exp-concavity conditions. In the online setting, the case of fixed functions $f_t = f$ seems rather restricted, however, and may in fact be handled by offline optimization methods. We therefore seek to loosen this requirement by replacing it by a stochastic condition on the distribution of the functions $f_t$. The relation between variance bounds like Theorem 1 and fast rates in the stochastic setting is studied in depth by Koolen, Grünwald, and Van Erven [19], who obtain fast rate results both in expectation and in probability. Here we provide a direct proof only for the expected regret, which allows a simplified analysis.

Suppose the functions $f_t$ are independent and identically distributed (i.i.d.), with common distribution $\mathbb{P}$. Then we say that the gradients satisfy the $(B, \beta)$-*Bernstein condition* with respect to the stochastic optimum $\boldsymbol{u}^* = \arg\min_{\boldsymbol{u} \in \mathcal{U}} \mathbb{E}_{f \sim \mathbb{P}}[f(\boldsymbol{u})]$ if

$$
(\boldsymbol{w} - \boldsymbol{u}^*)^{\intercal} \mathbb{E}_f\left[\nabla f(\boldsymbol{w}) \nabla f(\boldsymbol{w})^{\intercal}\right](\boldsymbol{w} - \boldsymbol{u}^*) \leq B\left((\boldsymbol{w} - \boldsymbol{u}^*)^{\intercal} \mathbb{E}_f[\nabla f(\boldsymbol{w})]\right)^{\beta} \qquad \text{for all } \boldsymbol{w} \in \mathcal{U}. \tag{4}
$$

This is an instance of the well-known Bernstein condition from offline statistical learning [1, 10], applied to the linearized excess loss $(\boldsymbol{w} - \boldsymbol{u}^*)^{\intercal} \nabla f(\boldsymbol{w})$. As shown in Appendix H, imposing the condition for the linearized excess loss is a weaker requirement than imposing it for the original excess loss $f(\boldsymbol{w}) - f(\boldsymbol{u}^*)$.

**Algorithm 1:** MetaGrad Master

---

**Input:** Grid of learning rates $\frac{1}{5DG} \geq \eta_1 \geq \eta_2 \geq \ldots$ with prior weights $\pi_1^{\eta_1}, \pi_1^{\eta_2}, \ldots$     ▷ *As in* (8)

1: **for** $t = 1, 2, \ldots$ **do**
2:      Get prediction $\boldsymbol{w}_t^\eta \in \mathcal{U}$ of slave (Algorithm 2) for each $\eta$
3:      Play $\boldsymbol{w}_t = \frac{\sum_\eta \pi_t^\eta \eta \boldsymbol{w}_t^\eta}{\sum_\eta \pi_t^\eta \eta} \in \mathcal{U}$              ▷ *Tilted Exponentially Weighted Average*
4:      Observe gradient $\boldsymbol{g}_t = \nabla f_t(\boldsymbol{w}_t)$
5:      Update $\pi_{t+1}^\eta = \frac{\pi_t^\eta e^{-\alpha \ell_t^\eta(\boldsymbol{w}_t^\eta)}}{\sum_\eta \pi_t^\eta e^{-\alpha \ell_t^\eta(\boldsymbol{w}_t^\eta)}}$ for all $\eta$     ▷ *Exponential Weights with surrogate loss* (6)
6: **end for**

---

**Theorem 3.** *If the gradients satisfy the* $(B, \beta)$*-Bernstein condition for* $B > 0$ *and* $\beta \in (0, 1]$ *with respect to* $\boldsymbol{u}^* = \arg\min_{\boldsymbol{u} \in \mathcal{U}} \mathbb{E}_{f \sim \mathbb{P}}[f(\boldsymbol{u})]$*, then any method with regret bound* (1) *incurs expected regret* $\mathbb{E}[R_T^{\boldsymbol{u}^*}] = O\left((Bd \ln T)^{1/(2-\beta)} T^{(1-\beta)/(2-\beta)} + d \ln T\right)$.

For $\beta = 1$, the rate becomes $O(d \ln T)$, just like for fixed functions, and for smaller $\beta$ it is in between logarithmic and $O(\sqrt{dT})$. For instance, the hinge loss on the unit ball with i.i.d. data satisfies the Bernstein condition with $\beta = 1$, which implies an $O(d \ln T)$ rate. (See Appendix A.4.) It is common to add $\ell_2$-regularization to the hinge loss to make it strongly convex, but this example shows that that is not necessary to get logarithmic regret.

## 4 MetaGrad Algorithm

In this section we explain the two versions (full and diagonal) of the MetaGrad algorithm. We will make use of the following definitions:

$$
\begin{array}{cc}
\text{full} & \text{diag} \\
\boldsymbol{M}_t^{\text{full}} \coloneqq \boldsymbol{g}_t \boldsymbol{g}_t^\mathsf{T} & \boldsymbol{M}_t^{\text{diag}} \coloneqq \text{diag}(g_{t,1}^2, \ldots, g_{t,d}^2) \\
\alpha^{\text{full}} \coloneqq 1 & \alpha^{\text{diag}} \coloneqq 1/d.
\end{array}
\tag{5}
$$

Depending on context, $\boldsymbol{w}_t \in \mathcal{U}$ will refer to the full or diagonal MetaGrad prediction in round $t$. In the remainder we will drop the superscript from the letters above, which will always be clear from context.

MetaGrad will be defined by means of the following *surrogate loss* $\ell_t^\eta(\boldsymbol{u})$, which depends on a parameter $\eta > 0$ that trades off *regret* compared to $\boldsymbol{u}$ with the square of the scaled directional derivative towards $\boldsymbol{u}$ (full case) or its approximation (diag case):

$$
\ell_t^\eta(\boldsymbol{u}) \coloneqq -\eta(\boldsymbol{w}_t - \boldsymbol{u})^\mathsf{T} \boldsymbol{g}_t + \eta^2 (\boldsymbol{u} - \boldsymbol{w}_t)^\mathsf{T} \boldsymbol{M}_t (\boldsymbol{u} - \boldsymbol{w}_t).
\tag{6}
$$

Our surrogate loss consists of a linear and a quadratic part. Using the language of Orabona, Crammer, and Cesa-Bianchi [26], the data-dependent quadratic part causes a "time-varying regularizer" and Duchi, Hazan, and Singer [9] would call it "temporal adaptation of the proximal function". The sum of quadratic terms in our surrogate is what appears in the regret bound of Theorem 1.

The MetaGrad algorithm is a two-level hierarchical construction, displayed as Algorithms 1 (master algorithm that learns the learning rate) and 2 (sub-module, a copy running for each learning rate $\eta$ from a finite grid). Based on our analysis in the next section, we recommend using the grid in (8).

**Master** The task of the Master Algorithm 1 is to learn the empirically best learning rate $\eta$ (parameter of the surrogate loss $\ell_t^\eta$), which is notoriously difficult to track online because the regret is non-monotonic over rounds and may have multiple local minima as a function of $\eta$ (see [18] for a study in the expert setting). The standard technique is therefore to derive a monotonic upper bound on the regret and tune the learning rate optimally *for the bound*. In contrast, our approach, inspired by the approach for combinatorial games of Koolen and Van Erven [16, Section 4], is to have our master aggregate the predictions of a discrete grid of learning rates. Although we provide a formal analysis of the regret, the master algorithm does not depend on the outcome of this analysis, so any

---
**Algorithm 2:** MetaGrad Slave
---
**Input:** Learning rate $0 < \eta \leq \frac{1}{5DG}$, domain size $D > 0$

1:  $\boldsymbol{w}_1^\eta = \boldsymbol{0}$ and $\boldsymbol{\Sigma}_1^\eta = D^2 \boldsymbol{I}$

2:  **for** $t = 1, 2, \ldots$ **do**

3:     Issue $\boldsymbol{w}_t^\eta$ to master (Algorithm 1)

4:     Observe gradient $\boldsymbol{g}_t = \nabla f_t(\boldsymbol{w}_t)$             ▷ *Gradient at* master *point* $\boldsymbol{w}_t$

5:     Update $\boldsymbol{\Sigma}_{t+1}^\eta = \left( \frac{1}{D^2}\boldsymbol{I} + 2\eta^2 \sum_{s=1}^t \boldsymbol{M}_s \right)^{-1}$

   $$\widetilde{\boldsymbol{w}}_{t+1}^\eta = \boldsymbol{w}_t^\eta - \boldsymbol{\Sigma}_{t+1}^\eta \left( \eta \boldsymbol{g}_t + 2\eta^2 \boldsymbol{M}_t(\boldsymbol{w}_t^\eta - \boldsymbol{w}_t) \right)$$

   $$\boldsymbol{w}_{t+1}^\eta = \Pi_{\mathcal{U}}^{\boldsymbol{\Sigma}_{t+1}^\eta} \left( \widetilde{\boldsymbol{w}}_{t+1}^\eta \right) \text{ with projection } \Pi_{\mathcal{U}}^{\boldsymbol{\Sigma}}(\boldsymbol{w}) = \arg\min_{\boldsymbol{u}\in\mathcal{U}}(\boldsymbol{u} - \boldsymbol{w})^\intercal \boldsymbol{\Sigma}^{-1}(\boldsymbol{u} - \boldsymbol{w})$$

6:  **end for**
---
Implementation: For $\boldsymbol{M}_t = \boldsymbol{M}_t^{\text{diag}}$ only maintain diagonal of $\boldsymbol{\Sigma}_t^\eta$. For $\boldsymbol{M}_t = \boldsymbol{M}_t^{\text{full}}$ use rank-one update $\boldsymbol{\Sigma}_{t+1}^\eta = \boldsymbol{\Sigma}_t^\eta - \frac{2\eta^2 \boldsymbol{\Sigma}_t^\eta \boldsymbol{g}_t \boldsymbol{g}_t^\intercal \boldsymbol{\Sigma}_t^\eta}{1 + 2\eta^2 \boldsymbol{g}_t^\intercal \boldsymbol{\Sigma}_t^\eta \boldsymbol{g}_t}$ and simplify $\widetilde{\boldsymbol{w}}_{t+1}^\eta = \boldsymbol{w}_t^\eta - \eta\boldsymbol{\Sigma}_{t+1}^\eta \boldsymbol{g}_t \left(1 + 2\eta \boldsymbol{g}_t^\intercal(\boldsymbol{w}_t^\eta - \boldsymbol{w}_t)\right)$.
---

slack in our bounds does not feed back into the algorithm. The master is in fact very similar to the well-known exponential weights method (line 5), run on the surrogate losses, except that in the predictions the weights of the slaves are *tilted* by their learning rates (line 3), having the effect of giving a larger weight to larger $\eta$. The internal parameter $\alpha$ is set to $\alpha^{\text{full}}$ from (5) for the full version of the algorithm, and to $\alpha^{\text{diag}}$ for the diagonal version.

**Slaves** The role of the Slave Algorithm 2 is to guarantee small surrogate regret for a fixed learning rate $\eta$. We consider two versions, corresponding to whether we take rank-one or diagonal matrices $\boldsymbol{M}_t$ (see (5)) in the surrogate (6). The first version maintains a *full* $d \times d$ covariance matrix and has the best regret bound. The second version uses only *diagonal* matrices (with $d$ non-zero entries), thus trading off a weaker bound with a better run-time in high dimensions. Algorithm 2 presents the update equations in a computationally efficient form. Their intuitive motivation is given in the proof of Lemma 5, where we show that the standard exponential weights method with Gaussian prior and surrogate losses $\ell_t^\eta(\boldsymbol{u})$ yields Gaussian posterior with mean $\boldsymbol{w}_t^\eta$ and covariance matrix $\boldsymbol{\Sigma}_t^\eta$. The full version of MetaGrad is closely related to the Online Newton Step algorithm [14] running on the original losses $f_t$: the differences are that each Slave receives the Master's gradients $\boldsymbol{g}_t = \nabla f_t(\boldsymbol{w}_t)$ instead of its own $\nabla f_t(\boldsymbol{w}_t^\eta)$, and that an additional term $2\eta^2 \boldsymbol{M}_t(\boldsymbol{w}_t^\eta - \boldsymbol{w}_t)$ in line 5 adjusts for the difference between the Slave's parameters $\boldsymbol{w}_t^\eta$ and the Master's parameters $\boldsymbol{w}_t$. MetaGrad is therefore a bona fide first-order algorithm that only accesses $f_t$ through $\boldsymbol{g}_t$. We also note that we have chosen the Mirror Descent version that iteratively updates and projects (see line 5). One might alternatively consider the Lazy Projection version (as in [34, 23, 32]) that forgets past projections when updating on new data. Since projections are typically computationally expensive, we have opted for the Mirror Descent version, which we expect to project less often, since a projected point seems less likely to update to a point outside of the domain than an unprojected point.

**Total run time** As mentioned, the running time is dominated by the slaves. Ignoring the projection, a slave with full covariance matrix takes $O(d^2)$ time to update, while slaves with diagonal covariance matrix take $O(d)$ time. If there are $m$ slaves, this makes the overall computational effort respectively $O(md^2)$ and $O(md)$, both in time per round and in memory. Our analysis below indicates that $m = 1 + \lceil \frac{1}{2} \log_2 T \rceil$ slaves suffice, so $m \leq 16$ as long as $T \leq 10^9$. In addition, each slave may incur the cost of a projection, which depends on the shape of the domain $\mathcal{U}$. To get a sense for the projection cost we consider a typical example. For the Euclidean ball a diagonal projection can be performed using a few iterations of Newton's method to get the desired precision. Each such iteration costs $O(d)$ time. This is generally considered affordable. For full projections the story is starkly different. We typically reduce to the diagonal case by a basis transformation, which takes $O(d^3)$ to compute using SVD. Hence here the projection dwarfs the other run time by an order of magnitude. We refer to [9] for examples of how to compute projections for various domains $\mathcal{U}$. Finally, we remark that a potential speed-up is possible by running the slaves in parallel.

# 5 Analysis

We conduct the analysis in three parts. We first discuss the master, then the slaves and finally their composition. The idea is the following. The master guarantees for all $\eta$ simultaneously that

$$0 \;=\; \sum_{t=1}^{T} \ell_t^\eta(\boldsymbol{w}_t) \;\leq\; \sum_{t=1}^{T} \ell_t^\eta(\boldsymbol{w}_t^\eta) + \text{master regret compared to } \eta\text{-slave.} \tag{7a}$$

Then each $\eta$-slave takes care of learning $\boldsymbol{u}$, with regret $O(d \ln T)$:

$$\sum_{t=1}^{T} \ell_t^\eta(\boldsymbol{w}_t^\eta) \;\leq\; \sum_{t=1}^{T} \ell_t^\eta(\boldsymbol{u}) + \eta\text{-slave regret compared to } \boldsymbol{u}. \tag{7b}$$

These two statements combine to

$$\eta \sum_{t=1}^{T} (\boldsymbol{w}_t - \boldsymbol{u})^\mathsf{T} \boldsymbol{g}_t - \eta^2 V_T^{\boldsymbol{u}} \;=\; - \sum_{t=1}^{T} \ell_t^\eta(\boldsymbol{u}) \;\leq\; \text{sum of regrets above} \tag{7c}$$

and the overall result follows by optimizing $\eta$.

## 5.1 Master

To show that we can aggregate the slave predictions, we consider the potential $\Phi_T \;:=\; \sum_\eta \pi_1^\eta e^{-\alpha \sum_{t=1}^{T} \ell_t^\eta(\boldsymbol{w}_t^\eta)}$. In Appendix B, we bound the last factor $e^{-\alpha \ell_T^\eta(\boldsymbol{w}_T^\eta)}$ above by its tangent at $\boldsymbol{w}_T^\eta = \boldsymbol{w}_T$ and obtain an objective that can be shown to be equal to $\Phi_{T-1}$ regardless of the gradient $\boldsymbol{g}_T$ if $\boldsymbol{w}_T$ is chosen according to the Master algorithm. It follows that the potential is non-increasing:

**Lemma 4** (Master combines slaves). *The Master Algorithm guarantees* $1 = \Phi_0 \geq \Phi_1 \geq \ldots \geq \Phi_T$.

As $0 \leq -\frac{1}{\alpha} \ln \Phi_T \leq \sum_{t=1}^{T} \ell_t^\eta(\boldsymbol{w}_t^\eta) + \frac{-1}{\alpha} \ln \pi_1^\eta$, this implements step (7a) of our overall proof strategy, with master regret $\frac{-1}{\alpha} \ln \pi_1^\eta$. We further remark that we may view our potential function $\Phi_T$ as a *game-theoretic supermartingale* in the sense of Chernov, Kalnishkan, Zhdanov, and Vovk [5], and this lemma as establishing that the MetaGrad Master is the corresponding *defensive forecasting* strategy.

## 5.2 Slaves

Next we implement step (7b), which requires proving an $O(d \ln T)$ regret bound in terms of the surrogate loss for each MetaGrad slave. In the full case, the surrogate loss is jointly exp-concave, and in light of the analysis of ONS by Hazan, Agarwal, and Kale [14] such a result is not surprising. For the diagonal case, the surrogate loss lacks joint exp-concavity, but we can use exp-concavity in each direction separately, and verify that the projections that tie the dimensions together do not cause any trouble. In Appendix C we analyze both cases simultaneously, and obtain the following bound on the regret:

**Lemma 5** (Surrogate regret bound). *For* $0 < \eta \leq \frac{1}{5DG}$, *let* $\ell_t^\eta(\boldsymbol{u})$ *be the surrogate losses as defined in* (6) *(either the full or the diagonal version). Then the regret of Slave Algorithm 2 is bounded by*

$$\sum_{t=1}^{T} \ell_t^\eta(\boldsymbol{w}_t^\eta) \leq \sum_{t=1}^{T} \ell_t^\eta(\boldsymbol{u}) + \frac{1}{2D^2} \|\boldsymbol{u}\|^2 + \frac{1}{2} \ln \det \left( \boldsymbol{I} + 2\eta^2 D^2 \sum_{t=1}^{T} \boldsymbol{M}_t \right) \qquad \text{for all } \boldsymbol{u} \in \mathcal{U}.$$

## 5.3 Composition

To complete the analysis of MetaGrad, we first put the regret bounds for the master and slaves together as in (7c). We then discuss how to choose the grid of $\eta$s, and optimize $\eta$ over this grid to get our main result. Proofs are postponed to Appendix D.

**Theorem 6** (Grid point regret). *The full and diagonal versions of MetaGrad, with corresponding definitions from* (2) *and* (5), *guarantee that, for any grid point* $\eta$ *with prior weight* $\pi_1^\eta$,

$$\tilde{R}_T^{\boldsymbol{u}} \;\leq\; \eta V_T^{\boldsymbol{u}} + \frac{\frac{1}{2D^2} \|\boldsymbol{u}\|^2 - \frac{1}{\alpha} \ln \pi_1^\eta + \frac{1}{2} \ln \det \left( \boldsymbol{I} + 2\eta^2 D^2 \sum_{t=1}^{T} \boldsymbol{M}_t \right)}{\eta} \qquad \text{for all } \boldsymbol{u} \in \mathcal{U}.$$

**Grid** We now specify the grid points and corresponding prior. Theorem 6 above implies that any two $\eta$ that are within a constant factor of each other will guarantee the same bound up to essentially the same constant factor. We therefore choose an exponentially spaced grid with a heavy tailed prior (see Appendix E):

$$\eta_i := \frac{2^{-i}}{5DG} \qquad \text{and} \qquad \pi_1^{\eta_i} := \frac{C}{(i+1)(i+2)} \qquad \text{for } i = 0, 1, 2, \ldots, \lceil \tfrac{1}{2} \log_2 T \rceil, \qquad (8)$$

with normalization $C = 1 + 1/(1 + \lceil \tfrac{1}{2} \log_2 T \rceil)$. At the cost of a worse constant factor in the bounds, the number of slaves can be reduced by using a larger spacing factor, or by omitting some of the smallest learning rates. The net effect of (8) is that, for any $\eta \in [\frac{1}{5DG\sqrt{T}}, \frac{2}{5DG}]$ there is an $\eta_i \in [\tfrac{1}{2}\eta, \eta]$, for which $-\ln \pi_1^{\eta_i} \le 2\ln(i+2) = O(\ln\ln(1/\eta_i)) = O(\ln\ln(1/\eta))$. As these costs are independent of $T$, our regret guarantees still hold if the grid (8) is instantiated with $T$ replaced by any upper bound.

The final step is to apply Theorem 6 to this grid, and to properly select the learning rate $\eta_i$ in the bound. This leads to our main result:

**Theorem 7** (MetaGrad Regret Bound). *Let* $\boldsymbol{S}_T = \sum_{t=1}^T \boldsymbol{M}_t$ *and* $V_{T,i}^{\boldsymbol{u}} = \sum_{t=1}^T (u_i - w_{t,i})^2 g_{t,i}^2$. *Then the regret of MetaGrad, with corresponding definitions from* (2) *and* (5) *and with grid and prior as in* (8)*, is bounded by*

$$\tilde{R}_T^{\boldsymbol{u}} \le \sqrt{8V_T^{\boldsymbol{u}}\left(\frac{1}{D^2}\|\boldsymbol{u}\|^2 + \Xi_T + \frac{1}{\alpha}C_T\right)} + 5DG\left(\frac{1}{D^2}\|\boldsymbol{u}\|^2 + \Xi_T + \frac{1}{\alpha}C_T\right) \qquad \text{for all } \boldsymbol{u} \in \mathcal{U},$$

*where*

$$\Xi_T \le \min\left\{\ln\det\left(\boldsymbol{I} + \frac{D^2\operatorname{rk}(\boldsymbol{S}_T)}{V_T^{\boldsymbol{u}}}\boldsymbol{S}_T\right), \operatorname{rk}(\boldsymbol{S}_T)\ln\left(\frac{D^2}{V_T^{\boldsymbol{u}}}\sum_{t=1}^T \|\boldsymbol{g}_t\|^2\right)\right\} = O(d\ln(D^2 G^2 T))$$

*for the full version of the algorithm,*

$$\Xi_T = \sum_{i=1}^d \ln\left(\frac{D^2 \sum_{t=1}^T g_{t,i}^2}{V_{T,i}^{\boldsymbol{u}}}\right) = O(d\ln(D^2 G^2 T))$$

*for the diagonal version, and* $C_T = 4\ln\left(3 + \tfrac{1}{2}\log_2 T\right) = O(\ln\ln T)$ *in both cases. Moreover, for both versions of the algorithm, the regret is simultaneously bounded by*

$$\tilde{R}_T^{\boldsymbol{u}} \le \sqrt{8D^2\left(\sum_{t=1}^T \|\boldsymbol{g}_t\|^2\right)\left(\frac{1}{D^2}\|\boldsymbol{u}\|^2 + \frac{1}{\alpha}C_T\right)} + 5DG\left(\frac{1}{D^2}\|\boldsymbol{u}\|^2 + \frac{1}{\alpha}C_T\right) \qquad \text{for all } \boldsymbol{u} \in \mathcal{U}.$$

These two bounds together show that the full version of MetaGrad achieves the new adaptive guarantee of Theorem 1. The diagonal version behaves like running the full version separately per dimension, but with a single shared learning rate.

## 6 Discussion and Future Work

One may consider extending MetaGrad in various directions. In particular it would be interesting to speed up the method in high dimensions, for instance by sketching [20]. A broader question is to identify and be adaptive to more types of easy functions that are of practical interest. One may suspect there to be a price (in regret overhead and in computation) for broader adaptivity, but based on our results for MetaGrad it does not seem like we are already approaching the point where this price is no longer worth paying.

**Acknowledgments** We would like to thank Haipeng Luo and the anonymous reviewers (in particular Reviewer 6) for valuable comments. Koolen acknowledges support by the Netherlands Organization for Scientific Research (NWO, Veni grant 639.021.439).

## Footnotes

[1]If $f_t$ is not differentiable at $\boldsymbol{w}_t$, any choice of subgradient $\boldsymbol{g}_t \in \partial f_t(\boldsymbol{w}_t)$ is allowed.

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
