[Supplementary Material]



(a) Offline: $f_t(u) = |u - 1/4|$

(b) Stochastic Online: $f_t(u) = |u - x_t|$ where $x_t = \pm\frac{1}{2}$ i.i.d. with probabilities 0.4 and 0.6.

Figure 1: Examples of fast rates on functions without curvature. MetaGrad incurs logarithmic regret $O(\ln T)$, while AdaGrad incurs $O(\sqrt{T})$ regret, matching its bound.

## A Extra Material Related to Section 3

In this section we gather extra material related to the fast rate examples from Section 3. We first provide simulations. Then we present the proofs of Theorems 2 and 3. And finally we give an example in which the unregularized hinge loss satisfies the Bernstein condition.

### A.1 Simulations: Logarithmic Regret without Curvature

We provide two simple simulation examples to illustrate the sufficient conditions from Theorems 2 and 3, and to show that such fast rates are not automatically obtained by previous methods for general functions. Both our examples are one-dimensional (so the full and diagonal algorithms coincide), and have a stable optimum (that good algorithms will converge to); yet the functions are based on absolute values, which are neither strongly convex nor smooth, so the gradient norms do not vanish near the optimum. As our baseline we include AdaGrad [9], because it is commonly used in practice [22, 27] and because, in the one-dimensional case, it implements GD with an adaptive tuning of the learning rate that is applicable to general convex functions.

In the first example, we consider offline convex optimization of the fixed function $f_t(u) \equiv f(u) = |u - \frac{1}{4}|$, which satisfies (3), because it is convex. In the second example, we look at stochastic optimization with convex functions $f_t(u) = |u - x_t|$, where the outcomes $x_t = \pm\frac{1}{2}$ are chosen i.i.d. with probabilities 0.4 and 0.6. These probabilities satisfy (4) with $\beta = 1$. Their values are by no means essential, as long we avoid the worst case where the probabilities are equal.

Figure 1 graphs the results. We see that in both cases the regret of AdaGrad follows its $O(\sqrt{T})$ bound, while MetaGrad achieves an $O(\ln T)$ rate, as predicted by Theorems 2 and 3. This shows that MetaGrad achieves a type of adaptivity that is not achieved by AdaGrad. We should be careful in extending this conclusion to higher dimensions, though: whereas (the diagonal version of) AdaGrad uses a separate learning rate per dimension, MetaGrad shares learning rates between dimensions (unless we run a separate copy of MetaGrad per dimension, as suggested in the related work section).

### A.2 Proof of Theorem 2

*Proof.* By (3), applied with $\boldsymbol{w} = \boldsymbol{w}_t$, and Theorem 1, there exists a $C > 0$ (depending on $a$) such that, for all sufficiently large $T$,

$$R_T^{\boldsymbol{u}} \le a\tilde{R}_T^{\boldsymbol{u}} - bV_T^{\boldsymbol{u}} \le C\sqrt{V_T^{\boldsymbol{u}} \, d \ln T} + Cd \ln T - bV_T^{\boldsymbol{u}}$$

$$\leq \frac{\gamma}{2}CV_T^{\boldsymbol{u}} + \left(\frac{1}{2\gamma}+1\right)Cd\ln T - bV_T^{\boldsymbol{u}} \qquad \text{for all } \gamma > 0,$$

where the last inequality is based on $\sqrt{xy} = \min_{\gamma>0}\frac{\gamma}{2}x + \frac{y}{2\gamma}$ for all $x, y > 0$. The result follows upon taking $\gamma = \frac{2b}{C}$. $\qquad\square$

### A.3  Proof of Theorem 3

*Proof.* Abbreviate $\tilde{r}_t^{\boldsymbol{u}} = (\boldsymbol{w}_t - \boldsymbol{u})^\mathsf{T}\boldsymbol{g}_t$. Then, by (1), Jensen's inequality and the Bernstein condition, there exists a constant $C > 0$ such that, for all sufficiently large $T$, the expected linearized regret is at most

$$\mathbb{E}\left[\tilde{R}_T^{\boldsymbol{u}^*}\right] \leq C\,\mathbb{E}\left[\sqrt{V_T^{\boldsymbol{u}^*}d\ln T}\right] + Cd\ln T \leq C\sqrt{\mathbb{E}\left[V_T^{\boldsymbol{u}^*}\right]d\ln T} + Cd\ln T$$

$$\leq C\sqrt{B\sum_{t=1}^{T}\left(\mathbb{E}\left[\tilde{r}_t^{\boldsymbol{u}^*}\right]\right)^\beta d\ln T} + Cd\ln T.$$

We will repeatedly use the fact that

$$x^\alpha y^{1-\alpha} = c_\alpha \inf_{\gamma>0}\left(\frac{x}{\gamma} + \gamma^{\frac{\alpha}{1-\alpha}}y\right) \qquad \text{for any } x, y \geq 0 \text{ and } \alpha \in (0,1), \tag{9}$$

where $c_\alpha = (1-\alpha)^{1-\alpha}\alpha^\alpha$. Applying this first with $\alpha = 1/2$, $x = Bd\ln T$ and $y = \sum_{t=1}^{T}\left(\mathbb{E}[\tilde{r}_t^{\boldsymbol{u}^*}]\right)^\beta$, we obtain

$$\sqrt{B\sum_{t=1}^{T}(\mathbb{E}[\tilde{r}_t^{\boldsymbol{u}^*}])^\beta\,d\ln T} \leq c_{1/2}\gamma_1\sum_{t=1}^{T}\left(\mathbb{E}[\tilde{r}_t^{\boldsymbol{u}^*}]\right)^\beta + \frac{c_{1/2}}{\gamma_1}Bd\ln T \qquad \text{for any } \gamma_1 > 0.$$

If $\beta = 1$, then $\sum_{t=1}^{T}\left(\mathbb{E}[\tilde{r}_t^{\boldsymbol{u}^*}]\right)^\beta = \mathbb{E}[\tilde{R}_T^{\boldsymbol{u}^*}]$ and the result follows by taking $\gamma_1 = \frac{1}{2Cc_{1/2}}$. Alternatively, if $\beta < 1$, then we apply (9) a second time, with $\alpha = \beta$, $x = \mathbb{E}[\tilde{r}_t^{\boldsymbol{u}^*}]$ and $y = 1$, to find that, for any $\gamma_2 > 0$,

$$\sqrt{B\sum_{t=1}^{T}(\mathbb{E}[\tilde{r}_t^{\boldsymbol{u}^*}])^\beta\,d\ln T} \leq c_\beta c_{1/2}\gamma_1\sum_{t=1}^{T}\left(\frac{\mathbb{E}[\tilde{r}_t^{\boldsymbol{u}^*}]}{\gamma_2} + \gamma_2^{\beta/(1-\beta)}\right) + \frac{c_{1/2}}{\gamma_1}Bd\ln T$$

$$= \frac{c_\beta c_{1/2}\gamma_1}{\gamma_2}\mathbb{E}[\tilde{R}_T^{\boldsymbol{u}^*}] + c_\beta c_{1/2}\gamma_1\gamma_2^{\beta/(1-\beta)}T + \frac{c_{1/2}}{\gamma_1}Bd\ln T.$$

Taking $\gamma_1 = \frac{\gamma_2}{2c_\beta c_{1/2}C}$, this yields

$$\mathbb{E}[\tilde{R}_T^{\boldsymbol{u}^*}] \leq \gamma_2^{1/(1-\beta)}T + \frac{4C^2c_{1/2}^2c_\beta Bd\ln T}{\gamma_2} + 2Cd\ln T.$$

We may optimize over $\gamma_2$ by a third application of (9), now with $x = 4C^2c_{1/2}^2c_\beta Bd\ln T$, $y = T$ and $\alpha = 1/(2-\beta)$, such that $\alpha/(1-\alpha) = 1/(1-\beta)$:

$$\mathbb{E}[\tilde{R}_T^{\boldsymbol{u}^*}] \leq \frac{1}{c_{1/(2-\beta)}}\left(4C^2c_{1/2}^2c_\beta Bd\ln T\right)^{1/(2-\beta)}T^{(1-\beta)/(2-\beta)} + 2Cd\ln T$$

$$= O\left((Bd\ln T)^{1/(2-\beta)}T^{(1-\beta)/(2-\beta)} + d\ln T\right),$$

which completes the proof. $\qquad\square$

### A.4  Unregularized Hinge Loss Example

As shown by Koolen, Grünwald, and Van Erven [19], the Bernstein condition is satisfied in the following classification task:

**Lemma 8** (Unregularized Hinge Loss Example). *Suppose that $(\boldsymbol{X}_1, Y_1), (\boldsymbol{X}_2, Y_2), \ldots$ are i.i.d. with $Y_t$ taking values in $\{-1, +1\}$, and let $f_t(\boldsymbol{u}) = \max\{0, 1 - Y_t \langle \boldsymbol{u}, \boldsymbol{X}_t \rangle\}$ be the hinge loss. Assume that both $\mathcal{U}$ and the domain for $\boldsymbol{X}_t$ are the $d$-dimensional unit ball. Then the $(B, \beta)$-Bernstein condition is satisfied with $\beta = 1$ and $B = \frac{2\lambda_{max}}{\|\boldsymbol{\mu}\|}$, where $\lambda_{max}$ is the maximum eigenvalue of $\mathbb{E}[\boldsymbol{X}\boldsymbol{X}^\mathsf{T}]$ and $\boldsymbol{\mu} = \mathbb{E}[Y\boldsymbol{X}]$, provided that $\|\boldsymbol{\mu}\| > 0$.*

*In particular, if $\boldsymbol{X}_t$ is uniformly distributed on the sphere and $Y_t = \text{sign}(\langle \bar{\boldsymbol{u}}, \boldsymbol{X}_t \rangle)$ is the noiseless classification of $\boldsymbol{X}_t$ according to the hyperplane with normal vector $\bar{\boldsymbol{u}}$, then $B \leq \frac{c}{\sqrt{d}}$ for some absolute constant $c > 0$.*

Thus the version of the Bernstein condition that implies an $O(d \ln T)$ rate is always satisfied for the hinge loss on the unit ball, except when $\|\boldsymbol{\mu}\| = 0$, which is very natural to exclude, because it implies that the expected hinge loss is $1$ (its maximal value) for all $\boldsymbol{u}$, so there is nothing to learn. It is common to add $\ell_2$-regularization to the hinge loss to make it strongly convex, but this example shows that that is not necessary to get logarithmic regret.

## B    Master Regret Bound (Proof of Lemma 4)

*Proof.* To prove Lemma 4, we start by bounding $e^{-\alpha \ell_t^\eta(\boldsymbol{w}_t^\eta)}$ by its tangent at $\boldsymbol{w}_t^\eta = \boldsymbol{w}_t$:

$$e^{-\alpha \ell_t^\eta(\boldsymbol{w}_t^\eta)} \leq 1 + \alpha\eta(\boldsymbol{w}_t - \boldsymbol{w}_t^\eta)^\mathsf{T} \boldsymbol{g}_t \qquad \text{for any } \eta \in (0, \tfrac{2}{3DG}]. \tag{10}$$

For the full case, where $\alpha = \alpha^{\text{full}} = 1$, this follows directly from the "prod bound" $e^{x - x^2} \leq 1 + x$ with $x = \eta(\boldsymbol{w}_t - \boldsymbol{w}_t^\eta)^\mathsf{T} \boldsymbol{g}_t$, which has previously been used in the prediction with expert advice setting [3, 11, 16] and holds for any $x \geq -2/3$. In the diagonal case, (10) does not hold with $\alpha = 1$, but it can be proved with $\alpha = \alpha^{\text{diag}} = 1/d$ by an application of Jensen's inequality combined with a separate prod bound per dimension:

$$e^{-\alpha \ell_t^\eta(\boldsymbol{w}_t^\eta)} = e^{\sum_i \frac{1}{d}\left(\eta(w_{t,i} - w_{t,i}^\eta)g_{t,i} - \eta^2(w_{t,i} - w_{t,i}^\eta)^2 g_{t,i}^2\right)} \overset{\text{Jensen}}{\leq} \sum_i \frac{1}{d} e^{\left(\eta(w_{t,i} - w_{t,i}^\eta)g_{t,i} - \eta^2(w_{t,i} - w_{t,i}^\eta)^2 g_{t,i}^2\right)}$$

$$\overset{\text{prod bound}}{\leq} \sum_i \frac{1}{d}\left(1 + \eta(w_{t,i} - w_{t,i}^\eta)g_{t,i}\right) = 1 + \alpha\eta(\boldsymbol{w}_t - \boldsymbol{w}_t^\eta)^\mathsf{T}\boldsymbol{g}_t.$$

We proceed to show that the potential $\Phi_T$ is non-increasing:

$$\Phi_{T+1} - \Phi_T = \sum_\eta \pi_1^\eta e^{-\alpha \sum_{t=1}^T \ell_t^\eta(\boldsymbol{w}_t^\eta)} \left(e^{-\alpha \ell_t^\eta(\boldsymbol{w}_{T+1}^\eta)} - 1\right)$$

$$\leq \sum_\eta \pi_1^\eta e^{-\alpha \sum_{t=1}^T \ell_t^\eta(\boldsymbol{w}_t^\eta)} \alpha\eta\left(\boldsymbol{w}_{T+1} - \boldsymbol{w}_{T+1}^\eta\right)^\mathsf{T} \boldsymbol{g}_{T+1} = 0,$$

where the inequality is the tangent bound (10), and the final equality is by definition of the master prediction (in fact, it can be taken as the motivation for the master's definition)

$$\boldsymbol{w}_{T+1} = \frac{\sum_\eta \pi_{T+1}^\eta \eta \boldsymbol{w}_{T+1}^\eta}{\sum_\eta \pi_{T+1}^\eta \eta} = \frac{\sum_\eta \pi_1^\eta e^{-\alpha \sum_{t=1}^T \ell_t^\eta(\boldsymbol{w}_t^\eta)} \eta \boldsymbol{w}_{T+1}^\eta}{\sum_\eta \pi_1^\eta e^{-\alpha \sum_{t=1}^T \ell_t^\eta(\boldsymbol{w}_t^\eta)} \eta}.$$

Since $\Phi_0 = 1$ is trivial, this completes the proof of the lemma.     $\square$

## C    Slave Regret Bound (Proof of Lemma 5)

*Proof.* For any distributions $P$ and $Q$ on $\mathbb{R}^d$, let $\text{KL}(P\|Q) = \mathbb{E}_P[\ln \frac{\mathrm{d}P}{\mathrm{d}Q}]$ denote the Kullback-Leibler divergence of $P$ from $Q$, and let $\boldsymbol{\mu}_P = \mathbb{E}_P[\boldsymbol{u}]$ denote the mean of $P$. In addition, let $\mathcal{N}(\boldsymbol{\mu}, \boldsymbol{\Sigma})$ denote a normal distribution with mean $\boldsymbol{\mu}$ and covariance matrix $\boldsymbol{\Sigma}$.

In round $t$, we play according to the mean of a multivariate Gaussian distribution $P_t$. In the first round, this is a normal distribution, which plays the role of a prior:

$$P_1 = \mathcal{N}(\boldsymbol{0}, D^2 \boldsymbol{I}).$$

Then we update using the exponential weights update, followed by a projection onto $\mathcal{P} = \{P : \boldsymbol{\mu}_P \in \mathcal{U}\}$, such that the mean stays in the allowed domain $\mathcal{U}$:

$$\mathrm{d}\tilde{P}_{t+1}(\boldsymbol{u}) = \frac{e^{-\ell_t^\eta(\boldsymbol{u})}\,\mathrm{d}P_t(\boldsymbol{u})}{\int_{\mathbb{R}^d} e^{-\ell_t^\eta(\boldsymbol{u}')}\,\mathrm{d}P_t(\boldsymbol{u}')}, \qquad\qquad P_{t+1} = \underset{P \in \mathcal{P}}{\arg\min}\,\mathrm{KL}(P\|\tilde{P}_t).$$

To see that Algorithm 2 implements this algorithm, we prove by induction that

$$P_t = \mathcal{N}(\boldsymbol{w}_t^\eta, \boldsymbol{\Sigma}_t^\eta).$$

For $t = 1$ this is clear, and if it holds for any $t$ then it can be verified by comparing densities that $\tilde{P}_{t+1} = \mathcal{N}(\tilde{\boldsymbol{w}}_{t+1}^\eta, \boldsymbol{\Sigma}_{t+1}^\eta)$. Since it is well-known that the KL-projection of a Gaussian $\mathcal{N}(\boldsymbol{\mu}, \boldsymbol{\Sigma})$ onto $\mathcal{P}$ is another Gaussian $\mathcal{N}(\boldsymbol{\nu}, \boldsymbol{\Sigma})$ with the same covariance matrix and mean $\boldsymbol{\nu} \in \mathcal{U}$ that minimizes $\frac{1}{2}(\boldsymbol{\nu} - \boldsymbol{\mu})^\mathsf{T}\boldsymbol{\Sigma}^{-1}(\boldsymbol{\nu} - \boldsymbol{\mu})$, it then follows that $P_{t+1} = \mathcal{N}(\boldsymbol{w}_{t+1}^\eta, \boldsymbol{\Sigma}_{t+1}^\eta)$. For completeness we provide a proof of this last result in Lemma 9 of Appendix F.

It now remains to bound the regret. Since $\mathcal{P}$ is convex, the Pythagorean inequality for Kullback-Leibler divergence implies that

$$\mathrm{KL}(Q\|\tilde{P}_{t+1}) \geq \mathrm{KL}(Q\|P_{t+1}) + \mathrm{KL}(P_{t+1}\|\tilde{P}_{t+1}) \geq \mathrm{KL}(Q\|P_{t+1})$$

for all $Q \in \mathcal{P}$. The following telescoping sum therefore gives us that

$$\mathrm{KL}(Q\|P_1) \geq \sum_{t=1}^T \mathrm{KL}(Q\|P_t) - \mathrm{KL}(Q\|P_{t+1}) \geq \sum_{t=1}^T \mathrm{KL}(Q\|P_t) - \mathrm{KL}(Q\|\tilde{P}_{t+1})$$

$$= \sum_{t=1}^T -\ln \underset{P_t}{\mathbb{E}}[e^{-\ell_t^\eta(\boldsymbol{u})}] - \underset{Q}{\mathbb{E}}[\ell_t^\eta(\boldsymbol{u})]. \tag{11}$$

This may be interpreted as a regret bound in the space of distributions, which we will now relate to our regret of interest. If $\boldsymbol{M}_t = \boldsymbol{M}_t^{\mathrm{full}}$, then Lemma 10 in Appendix G implies that

$$-\ln \underset{P_t}{\mathbb{E}}[e^{-\ell_t^\eta(\boldsymbol{u})}] \geq \ell_t^\eta(\boldsymbol{w}_t^\eta)$$

because $\boldsymbol{w}_t^\eta$ is the mean of $P_t$. Alternatively, if $\boldsymbol{M}_t = \boldsymbol{M}_t^{\mathrm{diag}}$ then $P_t$ has diagonal covariance $\boldsymbol{\Sigma}_t^\eta$, and we can use Lemma 10 again to draw the same conclusion.

To control $\mathbb{E}_Q[\ell_t^\eta(\boldsymbol{u})]$, we may restrict attention (without loss of generality by a standard maximum entropy argument) to normal distributions $Q = \mathcal{N}(\boldsymbol{\mu}, D^2\boldsymbol{\Sigma})$ with mean $\boldsymbol{\mu} \in \mathcal{U}$ and covariance $\boldsymbol{\Sigma} \succ \boldsymbol{0}$ (expressed relative to the prior variance $D^2$). Then, using the cyclic property and linearity of the trace,

$$\underset{Q}{\mathbb{E}}[\ell_t^\eta(\boldsymbol{u})] = -\eta(\boldsymbol{w}_t - \boldsymbol{\mu})^\mathsf{T}\boldsymbol{g}_t + \eta^2(\boldsymbol{w}_t^\mathsf{T}\boldsymbol{M}_t\boldsymbol{w}_t - 2\boldsymbol{\mu}^\mathsf{T}\boldsymbol{M}_t\boldsymbol{w}_t + \underset{Q}{\mathbb{E}}[\mathrm{tr}(\boldsymbol{u}^\mathsf{T}\boldsymbol{M}_t\boldsymbol{u})])$$

$$= -\eta(\boldsymbol{w}_t - \boldsymbol{\mu})^\mathsf{T}\boldsymbol{g}_t + \eta^2(\boldsymbol{w}_t^\mathsf{T}\boldsymbol{M}_t\boldsymbol{w}_t - 2\boldsymbol{\mu}^\mathsf{T}\boldsymbol{M}_t\boldsymbol{w}_t + \mathrm{tr}(\underset{Q}{\mathbb{E}}[\boldsymbol{u}\boldsymbol{u}^\mathsf{T}]\boldsymbol{M}_t))$$

$$= -\eta(\boldsymbol{w}_t - \boldsymbol{\mu})^\mathsf{T}\boldsymbol{g}_t + \eta^2(\boldsymbol{w}_t^\mathsf{T}\boldsymbol{M}_t\boldsymbol{w}_t - 2\boldsymbol{\mu}^\mathsf{T}\boldsymbol{M}_t\boldsymbol{w}_t + \mathrm{tr}((D^2\boldsymbol{\Sigma} + \boldsymbol{\mu}\boldsymbol{\mu}^\mathsf{T})\boldsymbol{M}_t))$$

$$= -\eta(\boldsymbol{w}_t - \boldsymbol{\mu})^\mathsf{T}\boldsymbol{g}_t + \eta^2\left((\boldsymbol{\mu} - \boldsymbol{w}_t)^\mathsf{T}\boldsymbol{M}_t(\boldsymbol{\mu} - \boldsymbol{w}_t) + D^2\,\mathrm{tr}(\boldsymbol{\Sigma}\boldsymbol{M}_t)\right)$$

$$= \ell_t^\eta(\boldsymbol{\mu}) + \eta^2 D^2\,\mathrm{tr}(\boldsymbol{\Sigma}\boldsymbol{M}_t).$$

Finally, it remains to work out

$$\mathrm{KL}(Q\|P_1) = \frac{1}{2D^2}\|\boldsymbol{\mu}\|^2 + \frac{1}{2}\left(-\ln\det\boldsymbol{\Sigma} + \mathrm{tr}(\boldsymbol{\Sigma}) - d\right).$$

We have now bounded all the pieces in (11). Putting them all together with the choice $\boldsymbol{\mu} = \boldsymbol{u}$ and optimizing the bound in $\boldsymbol{\Sigma}$ gives:

$$\sum_{t=1}^T \ell_t^\eta(\boldsymbol{w}_t^\eta) - \sum_{t=1}^T \ell_t^\eta(\boldsymbol{u}) \leq \frac{1}{2D^2}\|\boldsymbol{u}\|^2 + \frac{1}{2}\inf_{\boldsymbol{\Sigma}\succ\boldsymbol{0}}\left\{-\ln\det\boldsymbol{\Sigma} + \mathrm{tr}\left(\boldsymbol{\Sigma}\left(\boldsymbol{I} + 2\eta^2 D^2\sum_{t=1}^T \boldsymbol{M}_t\right)\right) - d\right\}$$

$$= \frac{1}{2D^2}\|\boldsymbol{u}\|^2 + \frac{1}{2}\ln\det\left(\boldsymbol{I} + 2\eta^2 D^2\sum_{t=1}^T \boldsymbol{M}_t\right), \tag{12}$$

where the minimum is attained at $\boldsymbol{\Sigma} = \left(\boldsymbol{I} + 2\eta^2 D^2\sum_{t=1}^T \boldsymbol{M}_t\right)^{-1}$. $\qquad\square$

# D Composition Proofs

Throughout this section we abbreviate $S_T = \sum_{t=1}^{T} M_t$.

## D.1 Proof of Theorem 6

*Proof.* We start with

$$0 \overset{\text{Lemma 4}}{\geq} \frac{1}{\alpha} \ln \Phi_T \geq \frac{1}{\alpha} \ln \pi_1^{\eta} - \sum_{t=1}^{T} \ell_t^{\eta}(\boldsymbol{w}_t^{\eta})$$

$$\overset{\text{Lemma 5}}{\geq} \frac{1}{\alpha} \ln \pi_1^{\eta} - \sum_{t=1}^{T} \ell_t^{\eta}(\boldsymbol{u}) - \frac{1}{2D^2} \|\boldsymbol{u}\|^2 - \frac{1}{2} \ln \det \left( \boldsymbol{I} + 2\eta^2 D^2 \boldsymbol{S}_T \right).$$

Now expanding the definition (6) of the surrogate losses we find

$$\eta \sum_{t=1}^{T} (\boldsymbol{w}_t - \boldsymbol{u})^{\mathsf{T}} \boldsymbol{g}_t \leq \frac{1}{2D^2} \|\boldsymbol{u}\|^2 - \frac{1}{\alpha} \ln \pi_1^{\eta} + \eta^2 V_T^{\boldsymbol{u}} + \frac{1}{2} \ln \det \left( \boldsymbol{I} + 2\eta^2 D^2 \boldsymbol{S}_T \right),$$

in which we may divide by $\eta$ to obtain the claim. $\square$

## D.2 Proof of Theorem 7

*Proof.* In principle we would like to directly select the $\eta$ that optimizes the regret bound from Theorem 6. But unfortunately we cannot tractably minimize that bound, since $\eta$ occurs in the $\ln \det$. To bring the $\eta$ out, we apply the variational form from (12) to Theorem 6 to obtain

$$\tilde{R}_T^{\boldsymbol{u}} \leq \inf_{\boldsymbol{\Sigma} \succ \boldsymbol{0}} \eta_i \left( V_T^{\boldsymbol{u}} + D^2 \operatorname{tr} \left( \boldsymbol{\Sigma} \boldsymbol{S}_T \right) \right) + \frac{\frac{1}{D^2} \|\boldsymbol{u}\|^2 - \frac{2}{\alpha} \ln \pi_1^{\eta_i} - \ln \det(\boldsymbol{\Sigma}) + \operatorname{tr}(\boldsymbol{\Sigma}) - d}{2\eta_i} \quad (13)$$

for all grid points $\eta_i$. This leads to an upper bound by plugging in a near-optimal choice for $\boldsymbol{\Sigma}$, which we choose as

$$\overset{\text{full}}{\boldsymbol{\Sigma} = (\boldsymbol{I} + c\boldsymbol{S}_T)^{-1}} \qquad \overset{\text{diag}}{\boldsymbol{\Sigma} = \frac{1}{D^2} \operatorname{diag}(V_{T,1}^{\boldsymbol{u}}, \ldots, V_{T,d}^{\boldsymbol{u}}) \boldsymbol{S}_T^{-1}},$$

where $c := \operatorname{rk}(\boldsymbol{S}_T) \left( \frac{D^2}{V_T^{\boldsymbol{u}}} - \frac{1}{\operatorname{tr}(\boldsymbol{S}_T)} \right)$ is non-negative because $V_T^{\boldsymbol{u}} \leq D^2 \sum_{t=1}^{T} \|\boldsymbol{g}_t\|^2 = D^2 \operatorname{tr}(\boldsymbol{S}_T)$ by Cauchy-Schwarz. We proceed to bound terms involving $\boldsymbol{\Sigma}$ above. In the diagonal case, we use that $D^2 \operatorname{tr}(\boldsymbol{\Sigma} \boldsymbol{S}_T) = V_T^{\boldsymbol{u}}$ and $\boldsymbol{\Sigma} \prec \boldsymbol{I}$ because $V_{T,i}^{\boldsymbol{u}} \leq D^2 \sum_{t=1}^{T} g_{t,i}^2$ by Cauchy-Schwarz. In the full case, we also have $\boldsymbol{\Sigma} \prec \boldsymbol{I}$. In addition, we observe that $\boldsymbol{S}_T$ and $\boldsymbol{\Sigma}$ share the same eigenbasis, so we may work in that basis. As $\boldsymbol{\Sigma} \boldsymbol{S}_T$ has $\operatorname{rk}(\boldsymbol{S}_T)$ non-zero eigenvalues, we may pull out a factor $\operatorname{rk}(\boldsymbol{S}_T)$ and replace the trace by a uniform average of the eigenvalues. Then Jensen's inequality for the concave function $x \mapsto \frac{x}{1+cx}$ for $x \geq 0$ gives

$$D^2 \operatorname{tr}(\boldsymbol{\Sigma} \boldsymbol{S}_T) \overset{\text{Jensen}}{\leq} \frac{D^2 \operatorname{tr}(\boldsymbol{S}_T)}{\left( 1 + \frac{c}{\operatorname{rk}(\boldsymbol{S}_T)} \operatorname{tr}(\boldsymbol{S}_T) \right)} = V_T^{\boldsymbol{u}}.$$

Thus, in both cases we have $D^2 \operatorname{tr}(\boldsymbol{\Sigma} \boldsymbol{S}_T) \leq V_T^{\boldsymbol{u}}$ and $\boldsymbol{\Sigma} \prec \boldsymbol{I}$, which implies that $\operatorname{tr}(\boldsymbol{\Sigma}) \leq \operatorname{tr}(\boldsymbol{I}) = d$ and that

$$\Xi_T := -\ln \det \boldsymbol{\Sigma} \geq 0.$$

Finally, by construction of the grid, for any $\eta \in [\frac{1}{5DG\sqrt{T}}, \frac{2}{5DG}]$ there exists a grid point $\eta_i \in [\frac{\eta}{2}, \eta]$, and the prior costs of this grid point satisfy

$$-\ln \pi_1^{\eta_i} \leq 2\ln(2+i) \leq 2\ln \left( 3 + \tfrac{1}{2} \log_2 T \right).$$

Plugging these bounds into (13) and abbreviating

$$A := \frac{1}{D^2} \|\boldsymbol{u}\|^2 + \frac{4}{\alpha} \ln \left( 3 + \tfrac{1}{2} \log_2 T \right) + \Xi_T \geq 4\ln 3,$$

we obtain

$$\tilde{R}_T^{\boldsymbol{u}} \;\leq\; 2\eta V_T^{\boldsymbol{u}} + \frac{A}{\eta}.$$

Subsequently tuning $\eta$ optimally as

$$\hat{\eta} \;=\; \sqrt{\frac{A}{2V_T^{\boldsymbol{u}}}} \geq \frac{\sqrt{2\ln 3}}{DG\sqrt{T}} \geq \frac{1}{5DG\sqrt{T}}$$

is allowed when $\hat{\eta} \leq \frac{2}{5DG}$, and gives $\tilde{R}_T^{\boldsymbol{u}} \leq \sqrt{8V_T^{\boldsymbol{u}}A}$. Alternatively, if $\hat{\eta} \geq \frac{2}{5DG}$, then we plug in $\eta = \frac{2}{5DG}$ and obtain $\tilde{R}_T^{\boldsymbol{u}} \leq \frac{4}{5DG}V_T^{\boldsymbol{u}} + \frac{5}{2}DGA \leq 5DGA$, where the second inequality follows from the constraint on $\hat{\eta}$. In both cases, we find that

$$\tilde{R}_T^{\boldsymbol{u}} \;\leq\; \sqrt{8V_T^{\boldsymbol{u}}A} + 5DGA,$$

which results in the first claim of the theorem upon observing that, for the full version of the algorithm, $\Xi_T \leq \mathrm{rk}(\boldsymbol{S}_T)\ln\left(\frac{D^2\,\mathrm{tr}(\boldsymbol{S}_T)}{V_T^{\boldsymbol{u}}}\right)$ by Jensen's inequality and $\Xi_T \leq \ln\det\left(\boldsymbol{I} + \frac{D^2\,\mathrm{rk}(\boldsymbol{S}_T)}{V_T^{\boldsymbol{u}}}\boldsymbol{S}_T\right)$ by monotonicity of $\ln\det$.

To prove the second claim, we instead take the comparator covariance $\boldsymbol{\Sigma} = \boldsymbol{I}$ equal to the prior covariance and again use $V_T^{\boldsymbol{u}} \leq D^2\,\mathrm{tr}(\boldsymbol{S}_T)$ to find

$$\tilde{R}_T^{\boldsymbol{u}} \;\leq\; \eta_i\left(V_T^{\boldsymbol{u}} + D^2\,\mathrm{tr}\left(\boldsymbol{S}_T\right)\right) + \frac{\frac{1}{D^2}\|\boldsymbol{u}\|^2 - \frac{2}{\alpha}\ln\pi_1^{\eta_i}}{2\eta_i} \;\leq\; 2\eta_i D^2\,\mathrm{tr}\left(\boldsymbol{S}_T\right) + \frac{\frac{1}{D^2}\|\boldsymbol{u}\|^2 - \frac{2}{\alpha}\ln\pi_1^{\eta_i}}{2\eta_i}$$

$$\leq\; 2\eta D^2\,\mathrm{tr}\left(\boldsymbol{S}_T\right) + \frac{\frac{1}{D^2}\|\boldsymbol{u}\|^2 + \frac{4}{\alpha}\ln\left(3 + \frac{1}{2}\log_2 T\right)}{\eta}$$

for all $\eta \in [\frac{1}{5DG\sqrt{T}}, \frac{2}{5DG}]$. Tuning $\eta$ as

$$\hat{\eta} = \sqrt{\frac{\frac{1}{D^2}\|\boldsymbol{u}\|^2 + \frac{4}{\alpha}\ln\left(3 + \frac{1}{2}\log_2 T\right)}{2D^2\,\mathrm{tr}\left(\boldsymbol{S}_T\right)}} \geq \sqrt{\frac{4\ln 3}{2D^2G^2T}} \geq \frac{1}{5DG\sqrt{T}}$$

is allowed when $\hat{\eta} \leq \frac{2}{5DG}$, and gives

$$\tilde{R}_T^{\boldsymbol{u}} \leq \sqrt{8D^2\,\mathrm{tr}\left(\boldsymbol{S}_T\right)\left(\frac{1}{D^2}\|\boldsymbol{u}\|^2 + \frac{4}{\alpha}\ln\left(3 + \frac{1}{2}\log_2 T\right)\right)}.$$

Alternatively, if $\hat{\eta} \geq \frac{2}{5DG}$, then we plug in $\eta = \frac{2}{5DG}$ and obtain

$$\tilde{R}_T^{\boldsymbol{u}} \;\leq\; \frac{4}{5DG}D^2\,\mathrm{tr}\left(\boldsymbol{S}_T\right) + \frac{5}{2}DG\left(\frac{1}{D^2}\|\boldsymbol{u}\|^2 + \frac{4}{\alpha}\ln\left(3 + \frac{1}{2}\log_2 T\right)\right)$$

$$\leq\; 5DG\left(\frac{1}{D^2}\|\boldsymbol{u}\|^2 + \frac{4}{\alpha}\ln\left(3 + \frac{1}{2}\log_2 T\right)\right),$$

where the second inequality follows from the constraint on $\hat{\eta}$. In both cases, the second claim of the theorem follows. $\qquad\square$

## E  Discussion of the Choice of Grid Points and Prior Weights

We now think about the choice of the grid and corresponding prior. Theorem 6 above implies that any two $\eta$ that are within a constant factor of each other will guarantee the same bound up to a constant factor. Since $\eta$ is a continuous parameter, this suggests choosing a prior that is approximately uniform for $\ln\eta$, which means it should have a density that looks like $1/\eta$. Although Theorem 6 does not show it, there is never any harm in taking too many grid points, because grid points that are very close together will behave as a single point with combined prior mass. If we disregard computation, we would therefore like to use the prior discussed by Chernov and Vovk [4], which is very close to uniform on $\ln\eta$ and has density

$$\pi(\eta) \;=\; \frac{C}{\eta\log_2^2\left(\frac{5}{2}DG\eta\right)},$$

where we include the factor $\frac{5}{2}DG$ to make the prior invariant under rescalings of the problem, and $C$ is a normalizing constant that makes the prior integrate to $1$. To adapt this prior to a discrete grid, we need to integrate this density between grid points and assign prior masses:

$$\pi_1^{\eta_i} := \int_{\eta_{i+1}}^{\eta_i} \pi(\eta)\, \mathrm{d}\eta = \left. \frac{C\ln(2)}{-\log_2(\frac{5}{2}DG\eta)} \right|_{\eta_{i+1}}^{\eta_i}.$$

For the exponentially spaced grid in (8), this evaluates to the prior weights $\pi_1^{\eta_i}$ specified there.

## F   Projection of Gaussians

It is well-known that the projection of a Gaussian onto the set of distributions with mean in the convex set $\mathcal{U}$ is also a Gaussian with the same covariance matrix. This result follows easily from, for instance, Theorems 1.8.5 and 1.8.2 of Ihara [15], but we include a short proof for completeness:

**Lemma 9.** *Let $\tilde{P}_t = \mathcal{N}(\boldsymbol{\mu}, \boldsymbol{\Sigma})$ be Gaussian and let $P_t = \arg\min_{P:\ \boldsymbol{\mu}_P \in \mathcal{U}} \mathrm{KL}(P \| \tilde{P}_t)$ be its projection onto the set of distributions with mean in $\mathcal{U}$. Then $P_t$ is also Gaussian with the same covariance matrix:*

$$P_t = \mathcal{N}(\boldsymbol{\nu}, \boldsymbol{\Sigma})$$

*for $\boldsymbol{\nu} \in \mathcal{U}$ that minimizes $\frac{1}{2}(\boldsymbol{\nu} - \boldsymbol{\mu})^\mathsf{T} \boldsymbol{\Sigma}^{-1}(\boldsymbol{\nu} - \boldsymbol{\mu})$.*

*Proof.* Let $P$ be an arbitrary distribution with mean $\boldsymbol{\nu} \in \mathcal{U}$, and let $R = \mathcal{N}(\boldsymbol{\nu}, \boldsymbol{\Sigma})$. Then by straight-forward algebra and nonnegativity of Kullback-Leibler divergence it can be verified that

$$\mathrm{KL}(P \| \tilde{P}_t) = \mathrm{KL}(P \| R) + \mathrm{KL}(R \| \tilde{P}_t) \geq \mathrm{KL}(R \| \tilde{P}_t).$$

Thus the minimum over all $P$ is achieved by a Gaussian with the same covariance matrix as $\tilde{P}_t$. It remains to find the mean of the projection, which is the $\boldsymbol{\nu} \in \mathcal{U}$ that minimizes

$$\mathrm{KL}(R \| \tilde{P}_t) = \frac{1}{2}(\boldsymbol{\nu} - \boldsymbol{\mu})^\mathsf{T} \boldsymbol{\Sigma}^{-1}(\boldsymbol{\nu} - \boldsymbol{\mu}),$$

as required. $\qquad\qquad\square$

## G   Gaussian Exp-concavity

Exp-concavity of $\ell_t^\eta(\boldsymbol{u})$ means that $\mathbb{E}\left[e^{-\ell_t^\eta(\boldsymbol{u})}\right] \leq e^{-\ell_t^\eta(\boldsymbol{\mu})}$ for any distribution on $\boldsymbol{u} \in \mathbb{R}^d$. Although this does not hold for general distributions with support outside of $\mathcal{U}$, it does hold if we restrict attention to certain types of Gaussians:

**Lemma 10** (Gaussian exp-concavity). *Let $0 < \eta \leq \frac{1}{5DG}$. Consider a Gaussian distribution with mean $\boldsymbol{\mu} \in \mathcal{U}$ and arbitrary covariance $\boldsymbol{\Sigma} \succ \mathbf{0}$ in the full case or diagonal $\boldsymbol{\Sigma} \succ \mathbf{0}$ in the diagonal case. Then*

$$\mathop{\mathbb{E}}_{\boldsymbol{u} \sim \mathcal{N}(\boldsymbol{\mu}, \boldsymbol{\Sigma})}\left[e^{-\ell_t^\eta(\boldsymbol{u})}\right] \leq e^{-\ell_t^\eta(\boldsymbol{\mu})}$$

*Proof.* We first consider the full case. Abbreviating $r := (\boldsymbol{w}_t - \boldsymbol{\mu})^\mathsf{T} \boldsymbol{g}_t$ and $s := (\boldsymbol{\mu} - \boldsymbol{u})^\mathsf{T} \boldsymbol{g}_t$, from the definition (6) of $\ell_t^\eta$ we get

$$\ell_t^\eta(\boldsymbol{\mu}) - \ell_t^\eta(\boldsymbol{u}) = \eta(\boldsymbol{\mu} - \boldsymbol{u})^\mathsf{T} \boldsymbol{g}_t - \eta^2 \left(2(\boldsymbol{\mu} - \boldsymbol{w}_t)^\mathsf{T} \boldsymbol{g}_t \boldsymbol{g}_t^\mathsf{T}(\boldsymbol{\mu} - \boldsymbol{u}) + (\boldsymbol{\mu} - \boldsymbol{u})^\mathsf{T} \boldsymbol{g}_t \boldsymbol{g}_t^\mathsf{T}(\boldsymbol{\mu} - \boldsymbol{u})\right)$$
$$= \eta s - \eta^2 \left(2rs + s^2\right).$$

Since $\boldsymbol{u} \sim \mathcal{N}(\boldsymbol{\mu}, \boldsymbol{\Sigma})$ implies $s \sim \mathcal{N}(0, v)$ with $v = \boldsymbol{g}_t^\mathsf{T} \boldsymbol{\Sigma} \boldsymbol{g}_t$, the claim collapses to

$$1 \geq \mathop{\mathbb{E}}_{\boldsymbol{u} \sim \mathcal{N}(\boldsymbol{\mu}, \boldsymbol{\Sigma})}\left[e^{\ell_t^\eta(\boldsymbol{\mu}) - \ell_t^\eta(\boldsymbol{u})}\right] = \mathop{\mathbb{E}}_{s \sim \mathcal{N}(0, v)}\left[e^{\eta s - \eta^2\left(2rs + s^2\right)}\right] = \frac{e^{\frac{\eta^2 v(1 - 2\eta r)^2}{2(1 + 2\eta^2 v)}}}{\sqrt{1 + 2\eta^2 v}},$$

which is equivalent to

$$(1 - 2\eta r)^2 \eta^2 v \leq \left(1 + 2\eta^2 v\right)\ln\left(1 + 2\eta^2 v\right).$$

The left-hand side is maximized over $r \in [-D^{\text{full}}G^{\text{full}}, D^{\text{full}}G^{\text{full}}]$ at $r = -D^{\text{full}}G^{\text{full}}$. So it suffices to establish

$$v(1 + 2\eta D^{\text{full}}G^{\text{full}})^2 \eta^2 \leq (1 + 2\eta^2 v) \ln(1 + 2\eta^2 v).$$

Now the right-hand is convex in $v$ and hence bounded below by its tangent at $v = 0$, which is $2\eta^2 v$. The proof is completed by observing that $(1 + 2\eta D^{\text{full}}G^{\text{full}})^2 \leq 2$ by the assumed bound on $\eta$.

It remains to consider the diagonal case. There the surrogate loss (6) is a sum over dimensions, say $\ell_t^\eta(\boldsymbol{u}) = \sum_{i=1}^d \ell_{t,i}^\eta(u_i)$. For a Gaussian with diagonal covariance matrix $\boldsymbol{\Sigma}$ the coordinates of $\boldsymbol{u}$ are independent, and hence

$$\mathop{\mathbb{E}}_{\boldsymbol{u} \sim \mathcal{N}(\boldsymbol{\mu}, \boldsymbol{\Sigma})} \left[ e^{-\ell_t^\eta(\boldsymbol{u})} \right] = \prod_{i=1}^d \mathop{\mathbb{E}}_{u_i \sim \mathcal{N}(\mu_i, \Sigma_{i,i})} \left[ e^{-\ell_{t,i}^\eta(u_i)} \right] \leq \prod_{i=1}^d e^{-\ell_{t,i}^\eta(\mu_i)} = e^{-\ell_t^\eta(\boldsymbol{\mu})},$$

where the inequality is the result for the full case applied to each dimension separately. $\square$

## H   Bernstein for Linearized Excess Loss

Let $f : \mathcal{U} \to \mathbb{R}$ be a convex function drawn from distribution $\mathbb{P}$ with stochastic optimum $\boldsymbol{u}^* = \arg\min_{\boldsymbol{u} \in \mathcal{U}} \mathbb{E}_{f \sim \mathbb{P}}[f(\boldsymbol{u})]$. For any $\boldsymbol{w} \in \mathcal{U}$, we now show that the Bernstein condition for the excess loss $X := f(\boldsymbol{w}) - f(\boldsymbol{u}^*)$ implies the Bernstein condition with the same exponent $\beta$ for the linearized excess loss $Y := (\boldsymbol{w} - \boldsymbol{u}^*)^\intercal \nabla f(\boldsymbol{w})$. These variables satisfy $Y \geq X$ by convexity of $f$ and $Y \leq C := D^{\text{full}}G^{\text{full}}$.

**Lemma 11.** *For $\beta \in (0, 1]$, let $X$ be a $(B, \beta)$-Bernstein random variable:*

$$\mathbb{E}[X^2] \leq B\,\mathbb{E}[X]^\beta.$$

*Then any bounded random variable $Y \leq C$ with $Y \geq X$ pointwise satisfies the $(B', \beta)$-Bernstein condition*

$$\mathbb{E}[Y^2] \leq B'\,\mathbb{E}[Y]^\beta$$

*for $B' = \max\left\{ B, \frac{2}{\beta}C^{2-\beta} \right\}$.*

*Proof.* For $\beta \in (0, 1)$ we will use the fact that

$$z^\beta = c_\beta \inf_{\gamma > 0} \left( \frac{z}{\gamma} + \gamma^{\frac{\beta}{1-\beta}} \right) \qquad \text{for any } z \geq 0,$$

with $c_\beta = (1 - \beta)^{1-\beta} \beta^\beta$. For $\gamma = \left( \frac{1-\beta}{\beta} \mathbb{E}[Y] \right)^{1-\beta}$ we therefore have

$$\mathbb{E}[X^2] - B'\,\mathbb{E}[X]^\beta \geq \mathbb{E}[X^2] - B'c_\beta \left( \frac{\mathbb{E}[X]}{\gamma} + \gamma^{\frac{\beta}{1-\beta}} \right) \geq \mathbb{E}[Y^2] - B'c_\beta \left( \frac{\mathbb{E}[Y]}{\gamma} + \gamma^{\frac{\beta}{1-\beta}} \right)$$

$$= \mathbb{E}[Y^2] - B'\,\mathbb{E}[Y]^\beta, \tag{14}$$

where the second inequality holds because $x^2 - c_\beta B' x / \gamma$ is a decreasing function of $x \leq C$ for $\gamma \leq \frac{c_\beta B'}{2C}$, which is satisfied by the choice of $B'$. This proves the lemma for $\beta \in (0, 1)$. The claim for $\beta = 1$ follows by taking the limit $\beta \to 1$ in (14). $\square$