[Reviews · NeurIPS 2016]

Reviewer 1

Summary

The paper deals with the question of online convex optimization (OCO). A new method named MetaGrad is designed, that is adaptive in the sense that it automatically catches the known fast regret rates for particular sets of convex functions (such as the largely studied strongly convex functions or exp-concave functions for instance, but also the less studied stochastic and nonstochastic functions without any curvature).

Qualitative Assessment

Though I am not familiar with the area of this work, I found the paper pleasant to read, and clearly written. The proposed method named MetaGrad is promising, and the derived upper bounds interesting. In particular, they raise the question of the avoidability of the extra d-term in such adaptive upper bounds, with respect to known upper bounds for methods designed for strongly convex functions, which is an important issue. I am not convinced that this extra term is unavoidable for adaptive general methods, as it is very large: the obtained upper bounds seem to be too pessimistic. Lower bounds for the regret of adaptive OCO procedures would definitely close the debate, giving challenging perspectives for future works.

Confidence in this Review

1-Less confident (might not have understood significant parts)


Reviewer 2

Summary

Online convex minimization is a fundamental problem in learning theory with various applications. Here, we are given a sequence of convex functions f1,...,fT in an online fashion and we seek to predict the minimizer of the sum so as to minimize the regret. This is a very well studied question with various guarantees known even when only given gradient information about the functions fi. For general functions this leads to a regret of sqrt(T) and for some special cases, we know better guarantees: O(log T) for strongly convex functions, O(d log T) for exp-convex functions etc. However, the algorithms that achieve the latter better guarantees are quite different and in some cases need to know bounds on the specific parameters of the functions (e.g., strong-convexity of the fi's, although this can now be avoided). The paper seeks to present a single algorithm which with a small overhead, achieves the right dependence on T for all the above cases. As such, this is a very nice research direction. Algorithm: The algorithm uses several techniques from previous works and while not very novel combines previous ideas in a clever way. The high level idea is to run several slave algorithms (that sort of correspond to using different parameter ranges for the base learning algorithm) and then use a master algorithm that combines the slave algorithms by using a modified exponential-weights method. This sort of approach of using two levels of learning algorithms is not particularly new to this approach, but the exact application is non-trivial. Writing: The paper is written reasonably well but the high-level description of the algorithm can use more work. Specifically, it is too vague in some very important places (e.g., "tilted by learning rates") and assumes too much familiarity with previous work ("adaptive regularizer", "temporal ...") that muddles the presentation. It is important to connect to previous works, but it will be more helpful for the reader if such terms can be explained and if not they do not seem to help much.

Qualitative Assessment

The problem and algorithm are nice but there are two important drawbacks: the new algorithm does not seem to cover (i.e., obtain better than O(sqrt(T)) guarantee) any interesting new cases; the bound for strongly-convex is worse than what is known already. Even with these drawbacks, I recommend the paper be accepted as the problem is quite nice and the approch could be useful in the future.

Confidence in this Review

2-Confident (read it all; understood it all reasonably well)


Reviewer 3

Summary

This paper considers online convex optimization and aims to construct algorithms with fast rates universally in broad classes of objective functions. The authors proposes the algorithms named MetaGrad and show that the regret of MetaGrad is bounded by the term depending on the variance in Theorems 1 and 7. Moreover, Theorem 1 is shown to guarantee that MetaGrad actually have fast rate without any manual tuning under the directional derivative condition or the Bernstein condition which covers various class of objective functions.

Qualitative Assessment

# Technical quality: The regret analysis for MetaGrad in the main manuscript and the corresponding proofs in the supplementary material are concise and intelligible. Simulations in the supplementary material compare MetaGrad and AdaGrad and clearly shows the effectiveness of MetaGrad for objective functions without curvature. However, It would be better if there were additional numerical experiments which compare MetaGrad and other state-of-the-art algorithms which are guaranteed to achieve fast rate for a specific function class such as strongly convex functions. # Novelty/originality: The proposed algorithm named MetaGrad is novel and its regret analysis guarantees the good performance for broad classes of objective functions. In particular, Theorems 2 and 3 give upper bounds of regrets for MetaGrad and can be applied for the class of objective functions without curvature and the upper bound is non-trivial from existing works. Actually, simple simulations in the supplemental material show the superiority of MetaGrad for AdaGrad. # Potential impact or usefulness: The optimization algorithms have often been proposed and analyzed per each class of objective functions, and hence, we had to carefully choose an appropriate algorithm depending on the class of objective functions if we hope good performance. This paper gives a way to overcome this issue by the proposed algorithms (MetaGrad) since those can be automatically tuned for wide range of classes of objective functions. Therefore, it is believed that this paper gives positive impact for the community of machine learning. # Clarity and presentation: The presentation is very clear and the logical structure is well-organized through the whole of the main manuscript and the supplementary material.

Confidence in this Review

2-Confident (read it all; understood it all reasonably well)


Reviewer 4

Summary

The paper introduces the MetaGrad algorithm for online convex optimization problems. MetaGrad uses a prediction with expert advice like algorithm to choose an optimal learning rate from a pool of candidate learning rates. Assuming some boundedness conditions on the domain and Lipschitz constants of the loss functions, the authors show that with O(log(T)) computation per step MetaGrad achieves regret R_T(u) = O[\sqrt{\sum ((u-w_t)g_t)^2 \log T} + d\log T]. In a few common practical settings (e.g. stochastic optimization), this implies logarithmic regret using only convexity without additional curvature assumptions (such as strong-convexity).

Qualitative Assessment

This paper is very well-written and contains a fascinating new form of regret bound that improves on worst-case analysis. The algorithm and analysis are explained in a lucid and thorough manner. The experiments were illustrative of the difference between log T and sqrt T, but not ultimately convincing of real-world applicability due to their simplicity. Since many of the applications that want to use MetaGrad will probably not satisfy either of the stated boundedness assumptions, I am curious about how dependent on correctly guessing these hyperparameters MetaGrad actually is in practice. However, given the potential strength of the theory, I feel that less than thorough experiments are a reasonable sacrifice.

Confidence in this Review

3-Expert (read the paper in detail, know the area, quite certain of my opinion)


Reviewer 5

Summary

This paper studies the problem of online convex optimization, and provides an algorithm which can work simultaneously for several types of loss functions, including general convex function, strongly convex function, and even some types of stochastic functions without any curvature. Previously, most works have different algorithms designed for different types of loss functions, or require the knowledge of some parameters characterizing the loss functions in order to set their learning rates appropriately. The idea of this paper is to maintain different learning rates simultaneously, by running different slave (or expert) algorithms with different learning rates. In addition, it uses a master algorithm to combine the decisions of the slaves into its own decision in an adaptive way, so that it can be almost as good as the slave with the right learning rate.

Qualitative Assessment

The results are nice and the presentation is clear in general. The idea of using different slaves (experts) with different learning rates and combining their decisions adaptively has appeared before in [17], for a related problem of combinatorial optimization with linear loss functions. For the convex optimization problem considered by this current paper, a natural approach is to replace linear functions with appropriate convex loss functions and let slaves run the online Newton step algorithm of [15]. Therefore, the full-version algorithm of this current paper, although nice, may be somewhat expected. To reduce the computational complexity, the authors propose a diagonal version of their algorithm, which requires some new ideas to make it work.

Confidence in this Review

2-Confident (read it all; understood it all reasonably well)


Reviewer 6

Summary

The paper introduces MetaGrad algorithm as an efficient adaptive method in online learning that guarantees state of the art regret bounds not only in worst case and arbitrary convex objective functions, but also small (logarithmic) regret bounds for (easier) large subclass of convex functions. The paper shows that MetaGrad demonstrates logarithmic regert bounds for at least two broad categories of objective convex functions: 1) functions satisfying "Directional Derivative Condition", and 2) functions with "Bernstein Stochastic Gradient". These categories also include some well-known subclasses such as exp-concave and strongly convex functions as well as unregularized hinge loss. The MetaGrad algorithm works with multiple learning rates at the same time via a master-slave architecture. Each Slave algorithm works with surrogate loss function and guarantees small regret for a fixed learning rate. Furthermore, the Master algorithm essentially learns the empirically best learning rate by collecting the predictions of all the slaves and plays an output.

Qualitative Assessment

The paper is very much well-written, well-organized and straight-forward to read. The proofs look sound and correct and the analysis is very thorough and complete in many aspects. The idea of using multiple learning rate at the same time in a hierarchical/master-slave architecture seems notably novel. The reviewer believes the paper will have a significant impact on the field and will further promote adaptive methods in online learning.

Confidence in this Review

2-Confident (read it all; understood it all reasonably well)